# Distribution of HPV Subtypes in Diverse Anogenital and Oral Samples from Women and Correlation of Infections with Neoplasia of the Cervix

**DOI:** 10.3390/cancers14133136

**Published:** 2022-06-26

**Authors:** Karen Bräutigam, Stefanie Meier, Sabina Meneder, Louisa Proppe, Katharina Stroschein, Stephan Polack, Frank Köster, Achim Rody, Sascha Baum

**Affiliations:** 1Department of Gynecology and Obstetrics, Campus Lübeck, University Medical Center Schleswig-Holstein, 23538 Lübeck, Germany; stefanie.meier@uksh.de (S.M.); sabina.meneder@uksh.de (S.M.); louisa.proppe@uksh.de (L.P.); katharina.stroschein@uksh.de (K.S.); stephan.polack@uksh.de (S.P.); frank.koester@uksh.de (F.K.); achim.rody@uksh.de (A.R.); 2Gynäkologie des MVZ Westpfalz, 67655 Kaiserslautern, Germany; ralphsbaum@googlemail.com

**Keywords:** HPV subtype distribution, anogenital and oral, cervical cancer, intraepithelial lesion, high-risk HPV

## Abstract

**Simple Summary:**

Human papilloma virus (HPV)-associated cancers and their precursors are increasing worldwide. The purpose of our study was to investigate HPV subtypes in anogenital and oral samples to analyze the distribution and frequency of high- and low-risk HPV in the cervix, vagina, vulva, anus, and oral cavity. Identification of special HPV subtypes for these areas may help to increase the prognostic value of HPV screening for early detection of precursors or cancers being highly HPV-attributable. HPV genotyping with the EUROArray HPV test was performed in 509 patient samples from our dysplasia consultation. In addition to the well described HPV, e.g., HPV 16 and HPV 31, we detected further HPV subtypes with higher incidences in the investigated areas (e.g., HPV 53 and HPV 73) which may be evident for precursors and cancers of all areas.

**Abstract:**

Background: Cancers and intraepithelial lesions of different anogenital areas as well as oral cancer are associated with human papilloma virus (HPV) infections. Methods: In this study cervical, vaginal, vulvar, anal, and oral samples were taken from 509 patients visiting our dysplasia consultation clinic. HPV genotyping was performed using the EUROArray HPV test. Results: Positivity of HR HPV was found in 60.4–64.3% of anogenital and 14.6% of oral samples. HPV 16 showed the highest incidence in all investigated areas. In cervical and vaginal samples HPV 31 was detected second most, while in vulvar, anal, and oral samples HPV 53 was the second most common subtype. HPV 18 was found lower in all areas, while HPV 51, HPV 52, and HPV 73 were detected higher than expected from published data. A good concordance between cervical, vaginal and vulvar samples was examined for most of the HPV. HR HPV infection was higher in cervical cancer (CC; 91.7%) and high-grade intraepithelial squamous lesions (HSIL; 93.9%) compared to low-grade SIL (LSIL; 69.6%) and normal samples (44.8%). Conclusion: In addition to the well described HPV subtypes, we found others with high incidences in the investigated areas which may be evident for HSIL and CC of those areas.

## 1. Introduction

Human papilloma virus (HPV)-associated cancers have increased worldwide in the last years. In addition to cervical cancer (CC), where more than 95% of all cases are attributable to HPV infections [1], other anogenital cancers, e.g., vaginal, vulvar and anal cancer are related to HPV infection comprising up to 75% [2], 50% [3,4] and over 90% [5] of the cases, respectively. Furthermore, HPV infection is highly associated with intraepithelial neoplasia (INs) of the different anogenital areas. According to the WHO classification from 2014, LSIL (low-grade squamous intraepithelial lesion) and HSIL (high-grade squamous intraepithelial lesion) are discriminated: LSIL described slight dysplasia, formerly known as C (cervical) IN 1, V (vulvar) IN 1, etc. Moderate to severe dysplasia (CIN 2-3, VIN 2-3, etc.) are summarized in HSIL [6]. An association of HPV infection and HSIL of anogenital areas are described for 85% and higher: CIN 2/3 up to 98.5% [7], anal IN 2/3 (AIN 2/3) up to 95% [5], vaginal IN 2/3 (VaIN 2/3) up to 96%, and VIN 2/3 up to 85% [3,4]. Additionally, accumulating oral HPV infections are the cause of the increasing incidence rates of oropharyngeal squamous cell carcinoma [8].

HPV-associated cancer develops after a persistent infection with oncogenic HPV genotypes of the HR subtypes including carcinogenic genotypes 16, 18, 31, 33, 35, 39, 45, 51, 52, 56, 58, and 59, and possibly carcinogenic genotypes HPV 68, 73, and 82, according to the definition of the International Agency for Research on Cancer (IARC) [9]. Furthermore, three genotypes are described as possibly carcinogenic (HPV 26, 53, and 66).

Requirement for the development of a precancerous disease or cancer is a persistent HPV infection. Here, the risk rises for an accidental integration of the viral DNA into the genome of the infected epithelial cells. An increased expression of the oncoproteins E6 and E7 of HR subtypes may lead to a malignant degeneration of infected epithelial cells through their influence on growth regulating intracellular signaling pathways [10,11,12].

Most HPV infections are transient and self-healing. Therefore, a distinct subtyping in line with medical monitoring is essential for the differentiation between persistent and transient infections of HR HPV. Though, both LR and HR HPV infections recover frequently without consequences, a higher risk must be taken to account when an HPV infection occurs, for both, men and women. New data showed that, e.g., every fourth infection with HPV 16 led to CIN3 or more [13].

In clinical routine diagnostics, detection of HPV is discriminated in only the two distinguished HPV 16 and HPV 18 while 12 HR HPV are pooled in the group of “HR HPV others” (31, 33, 35, 39, 45, 51, 52, 56, 58, 59, 66, 68). This study determines the frequencies and distributions of 18 HR and 12 LR HPV genotypes in samples of five anatomical areas including cervix, vagina, vulva, anus, and oral cavity and to correlate findings with patient clinical data.

## 2. Materials and Methods

### 2.1. Patient Samples and Ethical Considerations

This study was performed at the Department of Gynecology and Obstetrics of the University Medical Center Schleswig-Holstein, Campus Lübeck, Germany between September 2017 and February 2021. Overall, 540 women aged from 18 to 91 undergoing a colposcopy and/or a conization at the dysplasia consultation or visiting the outpatients were invited to participate in the study. The study was approved by the ethics committee of Lübeck (AZ 17-155). After recruitment, patients were interviewed about their anamnesis and elucidated by a physician. Further, patients’ data were collected by the in-house medical database. From the 540 patients 31 were excluded because of missing clinical information. For sampling, samples were taken of the five anatomical areas (cervix, vagina, vulva (between major and minor labia), anus (perianal), and oral cavity (cheek)) at the same time by a physician using flocked swabs in eNat^®^ (Guanidine-thiocyanate based) medium (Copan Italia, Brescia, Italy) and stored at room temperature. If available, information of the regular examination performed as colposcopy, cytology, and histology, were reported.

### 2.2. DNA Isolation of Epithelial Cells of the Swab Samples

The samples were proceeded in the oncological laboratory of the Department of Gynecology and Obstetrics of the University Medical Center Schleswig-Holstein, Campus Lübeck, Germany.

Total DNA was extracted from swabs using the QIAamp DNA mini kit (Qiagen, Hilden, Germany) according to manufacturer’s instructions. Supplementary to the standard protocol, samples were vortexed for 5 s after addition of proteinase K. Each time after pipetting AL buffer and ethanol, the samples were vortexed 3 times for 5 s. For DNA elution 50 μL of AE buffer were applied to the column and incubated for 5 min at room temperature before final centrifugation at 8000× *g* for 1 min. Isolated DNA were stored at −20 °C until further processing.

### 2.3. Detection of HPV Subtypes Using the EUROArray HPV Test

The EUROArray HPV test (EUROIMMUN Medizinische Labordiagnostika AG, Lübeck, Germany) is based on amplification and detection of the viral oncogenes E6/E7 via PCR and hybridization with immobilized DNA probes in a microarray system. By using subtype specific primer and probe systems, 30 anogenital HPV genotypes are detected and differentiated simultaneously in one multiplexed reaction; 18 HR HPV (16, 18, 26, 31, 33, 35, 39, 45, 51, 52, 53, 56, 58, 59, 66, 68, 73, 82) and 12 LR HPV (6, 11, 40, 42, 43, 44, 54, 61, 70, 72, 81, 89). The hybridization of the PCR product to the corresponding probe is detected using the EUROArrayScanner. The EUROArrayScan software subsequently evaluates all spot signals (relative fluorescent intensity) and generates a qualitative test result, i.e., detected/not detected based on HPV-type specific cut offs.

### 2.4. Statistics

The data were primarily analyzed descriptively. Phi coefficient was used to determine association for HPV positivity (or negativity) between different anatomical sites—i.e., cervix/vagina, cervix/vulva, cervix/anus, and cervix/mouth (oral)—for each HPV type or HPV type group (HR HPV and LR HPV).

Obesity, age, and number of pregnancies and children were correlated with HPV infection in a set of 477 patients where samples of all five areas were available. Additionally, use of contraceptives and hormones, both at the time of investigation or earlier were correlated with HPV positivity in 310 patients where data were available. Association of use of a hormonal intrauterine device (IUD) and existence of CC or CIN were analyzed by Fisher’s exact test.

The possibility of the agreement of analyzed subtypes occurring by chance within the different areas were determined by Cohen’s kappa coefficient (κ) with κ < 0 = poor, 0–0.20 = slight, 0.21–0.40 = fair, 0.41–0.60 = moderate, 0.61–0.80 = substantial, and 0.81–1.00 = (almost) perfect agreement. The chances of being positive at other sites while positive at cervix were analyzed by logistic regression and age adjusted odd ratios. A *p*-value ≤ 0.05 was considered statistically significant. Analyses were performed using R, version 4.0.2, [14]. Microsoft Excel 2013, and GraphPad Prism 6 for Windows (6.03, 2013, GraphPad Software, Inc., San Diego, CA, USA).

## 3. Results

### 3.1. Demographics of the Participants

In total, 509 women were included in the study and the results are based on data from all women unless stated otherwise. The participants’ ages ranged from 18 to 91 years (mean = 43.5 years (±14.7).

Almost half of the recruited women reported of having been pregnant two or more times (49.3%, 245/497) and 60.8% (302/497) had children. In total 49.7% (247/497) had one or two children while 11.1% (55/497) had three or more children. Smoking was reported from 40.1% (194/484) of the women. Furthermore, 18.9% of the women were severely overweight (adiposity; body mass index (BMI) > 30) (Table 1).

Current oral contraceptive (pill) use was reported by 28.8% (128/444) whereas 6.5% (28/433) reported using a hormonal IUD. A total of 10.9% (47/433) of the women reported using other hormone-based contraceptive methods. Information regarding HPV vaccination was sparse and insufficient for deep statistical analyses; 37 women of 181 (20.4%) reported being immunized. As expected, because in Germany the vaccination program started in 2007, the median age of vaccinated women was 28.3 ± 7.3 years younger than the median age of the non-vaccinated group (39.7 ± 10.9 years). Furthermore, the information regarding the sort of vaccine was incomplete. Complete vaccination was reported for 13 patients with one applied Gardasil 4, four Gardasil 9, and two Cervarix. For the others, neither vaccine type nor the frequency of vaccination was known. In total, within the 37 vaccinated patients ten had a normal cytology, ten PAP 3, one AIN 1, one CIN 1, eight CIN 2, and seven CIN 3. Of the vaccinated patients with diagnoses of CIN 1–3, all were HR HPV-positive. Of these HR HPV, six were HPV 16 or HPV 18 which would have been covered by all vaccines and ten were HR HPV subtypes covered only by Gardasil 9. Samples with PAP 3 or normal cytology showed less HPV 16, 18 or HR HPV subtypes. Of the 142 patients without vaccination, seven were diagnosed with CC, 51 with CIN 3, 14 with CIN 2, 13 with CIN 1, 18 with a conspicuous PAP, and 44 with PAP 2a/normal cytology. In six of seven CC samples, HPV 16 was analyzed and would have been covered by any vaccination type. In CIN 3 samples, 48 of 51 samples showed HR HPV 16, 18, 31, 33, 45, 52, or 58 and the patients would have benefitted from vaccination with Gardasil 9; and still 31 samples with HPV 16 or 18 would have benefitted from vaccination with either Gardasil 4 or Cervarix. Contribution of the seven HR HPV included in the nonavalent vaccination with Gardasil 9 in samples of non-vaccinated and vaccinated were shown in Table 2.

### 3.2. Cytological Examination

Among the 487 patients with cervical swab samples and cytopathological examination, the overall prevalence of precancerous lesions and CC was 37.0% (180/487). In detail, 4.7% (23/487) displayed low-grade neoplasia (CIN 1), 8.6% (42/487) showed CIN 2, 18.5% (90/487) CIN 3, and 5.1% (25/487) samples corresponded to carcinoma (Table 3). The age of the women presenting with precancerous lesions and carcinoma ranged between 19 and 79 years (mean 38.8 years).

Vaginal intraepithelial neoplasia (VaIN) and vaginal carcinoma (VaC) were detected in 1.8% (9/509); VaIN 1 in 0.8% (4/509), VaIN 2 in 0.2% (1/509) and VaIN 3 and VaC in 0.4% (2/509) of the 509 women with vaginal swab samples; (Table 3). The ages of the women with VaIN or VaC ranged from 22 to 85 years (mean 46.8 years).

The prevalence of intraepithelial neoplasia in the vulva (VIN) or vulvar carcinoma (VC) was 5.8% (29/502); VIN 1 0.4% (2/502), VIN 2 and VIN 3 each 1.2% (6/502) and VC 3.0% (15/502) (Table 3). The women presenting with vulvar lesions and carcinoma were between 28 and 91 years of age (mean 59.9 years).

Anal intraepithelial neoplasia (AIN) of grade 1 was detected in 0.4% (2/482) of the samples while AIN grade 3 was detected in 0.6% (3/482) of the samples. Women with AIN were between 30 to 50 years of age (mean 43.8 years). No anal carcinoma was detected (Table 3).

No oral intraepithelial neoplasia or carcinoma were detected.

### 3.3. Detection and Frequency of HPV Genotypes in Examined Areas

All 30 HPV genotypes were detected at varying frequencies.

Of the 487 cervical samples tested, 348 (71.5%) were HPV positive (HR, LR or both). HR HPV types were detected in 300 cervical samples (61.6%). Of those, 164 samples (54.7%) showed single HR HPV infections, while 136 (45.3%) had multiple HR HPV genotypes (Table 4). The most commonly detected HR HPV genotype was HPV 16 (45.7%), followed by HPV 31 (17.3%), HPV 52 (12.0%), HPV 53 (11.3%), and HPV 18 (9.0%) (Table 5). LR HPV were detected in 158 cervical samples (32.4%). Of those, 114 samples (72.2%) showed a single LR HPV infection, and 44 (27.8%) multiple LR HPV genotypes, with HPV 42 (29.1%) as the most common subtype, followed by HPV 54 (24.7%), HPV 40 (22.8%), HPV 6 (15.8%), and HPV 44 (11.4%) (Table 5).

Of the vaginal samples, 361 of 509 (70.9%) were HPV positive (HR, LR or both). HR HPV types were detected in 318 vaginal samples (62.5%). In 161 samples of HR HPV single genotype infections (50.6%) were verified, while 157 samples (49.4%) had multiple infecting HPV genotypes (Table 4). The most commonly detected HR HPV genotype was HPV 16 (50.0%), followed by HPV 31 (16.0%), HPV 53 (13.8%), HPV 52 (12.6%), and HPV 39 (9.4%) (Table 5). LR HPV were detected in 187 samples of 509 (36.7%). In 120 samples (64.2%) single LR HPV infections were shown, and 67 samples (35.8%) displayed multiple HPV genotypes, with HPV 42 as most frequent (30.0%), followed by HPV 54 (24.6%), HPV 6 (17.1%), HPV 40, and HPV 44 (16.0% each) (Table 5).

Of the vulvar samples 365 of 502 (72.7%) were HPV positive (HR, LR or both). HR HPV types were detected in 323 vulvar samples (64.3%). Single HR HPV genotype infections were shown in 182 samples (56.3%), and 141 (43.7%) had multiple HPV genotypes (Table 4). HPV 16 was the most frequent genotype of HR HPV (53.9%), followed by HPV 53 (17.7%), HPV 31 (15.8%), HPV 52 (10.8%), and HPV 51 (8.7%) (Table 5). LR HPV types were detected identified in 181 of 502 samples (36.1%). In total, 119 samples (65.7%) indicated a single LR HPV infection, and 62 (34.3%) multiple HPV genotypes. HPV 42 was determined most often (33.7%), followed by HPV 54 (26.5%), HPV 61 (18.2%), HPV 6 (17.7%), and HPV 44 (13.3%) (Table 5).

Of the anal samples, 344 of 482 (71.4%) were HPV positive (HR, LR or both). In total, 291 samples were positive for HR HPV (60.4%). Of those, 169 samples (58.1%) exhibited single genotype infection, while 122 samples had (41.9%) multiple HPV genotypes (Table 4).

The most commonly detected genotype of HR HPV was HPV 16 (54.0%), followed by HPV 53 (18.9%), HPV 31 (11.3%), HPV 73 (10.3%), and HPV 51 (9.6%) (Table 5). LR HPV were identified in 190 samples (39.4%) with 133 samples (70.0%) showing a single, and 57 (30.0%) multiple LR HPV infections. HPV 42 was the most frequent LR HPV (30.0%), followed by HPV 54 (27.9%), HPV 6 (22.6%), HPV 61 (16.3%), and HPV 44 (15.3%) (Table 5).

As expected, the prevalence of HPV and HR HPV infections were higher in women with precancerous lesions (CIN/VaIN/VIN ≥ 2) and carcinoma compared to women with normal cytology (Table 3). All women presenting with anal intraepithelial neoplasia were positive for HR HPV in their anal samples.

Oral HPV infections were less frequent than genital and anal HPV infections with 20 different HPV types being detected in total. In total, 97 of 507 oral samples (71.4%) were HPV DNA positive (HR, LR or both). HR HPV genotypes were detected in 74 samples (14.6%). Of those, 58 samples (78.4%) showed single HR HPV genotype infection, and 16 (21.6%) had multiple HPV genotypes (Table 4). The most commonly detected HR HPV genotypes were HPV 16 (79.7%), HPV 53 (14.9%), HPV 51 (9.5%), and HPV 31 and 73 (4.1% each) (Table 5). LR HPV genotypes were detected in 40 samples of 507 (7.9%). Here, 35 samples (70.0%) showed a single LR HPV infection, and 5 (30.0%) had 2 multiple HPV genotypes, with HPV 42 as the most common LR HPV (60.0%), followed by HPV 54 and HPV 6 (17.5% each), HPV 61 (10.0%), and HPV 89 (2.5%) (Table 5).

Looking for co-infections of both HR and LR HPV in the same anatomical areas, we found 109 (22.4%) in the cervix, 144 (28.3%) in the vagina, 139 (27.7%) in the vulva, 137 (28.4%) in the anus, and 17 (3.4%) in the oral cavity. HR HPV 16 showed the highest co-infection rates together with the LR HPV 42 with rising percentages from cervix (13.8%), vagina (14.6%), vulva (17.3%), anus (21.2%), to the oral cavity (29.4%). HR HPV 31 was analyzed most frequently in combination with the LR HPV 54 with decreasing percentages from cervix (8.3%), vagina (7.6%), vulva (7.2%), anus (3.7%), to the oral cavity (0%).

### 3.4. Concordance of HPV Detection among the Different Anatomical Sites

There was a medium to strong association in the detection of HPV, HR HPV and LR HPV between the cervix/vagina and cervix/vulva areas (Table 6). The phi-coefficient for HR HPV between the cervix/vagina and the cervix/vulva was 0.71 and 0.67, respectively. For LR HPV, phi coefficients were 0.59 and 0.55 between the cervix/vagina and the cervix/vulva, respectively.

Being HPV positive at the cervix extensively increased the chances of being positive at other sites: Age-adjusted odds ratio for cervical/vaginal association was 28.39 ([95% CI: 16.73–49.69] *p* = 1.4 × 10^−33^), for cervical/vulvar 18.10 ([95% CI: 10.93–30.71] *p* = 3.4 × 10^−28^), for cervical/anal 4.58 ([95% CI: 2.91–7.27] *p* = 6.3 × 10^−11^), and for cervical/oral 1.88 ([95% CI: 1.09–3.42] *p* = 0.03).

The concordance in detection for most of the individual HPV types was high. Determination of kappa coefficients was performed to estimate the agreement of detection of the most frequent HR HPV within the different areas (Figure 1). HPV 16 in cervical samples showed a substantial agreement to vaginal (87.5%, κ = 0.70) and vulvar samples (83.3%, κ = 0.61), and a moderate agreement to anal samples (78.4%, κ = 0.48). A substantial concordance for HPV 16 was seen between vaginal and vulvar samples, each of them with a moderate agreement to anal samples. Only slight agreements were observed to oral samples (κ = 0.11–0.17). An almost perfect concordance existed for HPV 31 between cervical and vaginal samples (97.3%, κ = 0.86), for HPV 51 between cervical and vulvar samples (98.1%, κ = 0.81) as well as for HPV 52 between cervical and vaginal (98.2%, κ = 0.87) and vaginal and vulvar samples (97.7%, κ = 0.81). Interestingly, a substantial concordance was proven for HPV 73 in all anogenital areas. We found no agreement between the HPV types detected in any of the anogenital areas compared to the types detected in the oral samples, except a moderate concordance for HPV 52 between cervix and oral samples.

### 3.5. Prevalence of HPV Subtypes in Cervical Intraepithelial Lesions and Cancer

In our study cohort, patients with vaginal, vulvar, and anal intraepithelial lesions and cancers were rare. Therefore, we analyzed HPV subtype prevalence exclusively for CIN and cervical cancer.

Among the 24 cervical cancer samples, 95.5% were positive for any HPV type while 91.7% were positive for HR HPV (Table 3 and Table 7). In total, 66.7% of the samples were positive for HPV 16. The second most common HR HPV were HPV 52, HPV 53, HPV 58, and HPV 73 (8.3% each; Table 7). Out of the 90 CIN 3 samples, 96.7% were HPV positive and 94.4% were HR HPV positive. The most common subtype was HPV 16 (48.9%), followed by HPV 31 (23.3%), HPV 52 (16.7%) and HPV 33 and HPV 45 (each 10%). Among the CIN 2 samples (n = 90), 97.6% were HPV positive and 92.9% were HR HPV positive. The most common subtype was HPV 16 with 33.3%, followed by HPV 31 with 23.8%, HPV 52 (21.4%), HPV 18, HPV 33, and HPV 73 (each 11.9%). In CIN 1 samples (n = 23), 73.9% were positive for any HPV and 69.6% were HR HPV positive. The most common subtype was HPV 16 with 21.7%, followed by HPV 31 with 13.0%, HPV 26, HPV 52, HPV 58, HPV 59, and HPV 68 (each 8.7%). In CIN 0/normal samples (CIN 0 referred to patients with biopsies taken (non-malignant), normal to patients without biopsy taken), 58.3% were positive for any HPV and 44.8% were positive for HR HPV subtypes. Moreover, in the CIN 0/normal samples the most common HR HPV was HPV 16 (18.1%) but with lower percentages compared to CIN 1–3 and CC. The second most common HR HPV was HPV 39 and HPV 53 (6.6% each), followed by HPV 31 (6.3%), HPV 18 and HPV 26 (5.2% each) (Table 7).

Considering the new nomenclature [6] we also analyzed the groups into HSIL and LSIL. In HSIL (CIN 2/3) samples, 97.0% were HPV positive and 93.9% were HR HPV positive. The most common subtype was HPV 16 with 43.9%, followed by HPV 31 with 23.5%, and HPV 52 (18.2%). In LSIL (CIN 1) samples, 73.9% were HPV positive and 69.6% HR HPV positive. Here, more different subtypes with lower percentages were investigated. The most common was still HPV 16 (21.7%), followed by HPV 31 (13.0%) and HPV 52 (8.7%) (Table 8). HPV, HR HPV, and HPV 16 positivity was higher at a statistically significant level in HSIL compared to LSIL (2.213 × 10^−5^, 2.150 × 10^−4^, 0.046).

### 3.6. HPV Association to Patients’ Data

HPV prevalence in the genital and anal areas decreased with age with the highest prevalence among women ≤39 years, while the oral HPV prevalence remained constant with minor fluctuations (Figure 2). In 477 patients, age was the only factor associated with an increased chance of being HPV positive (*p* = 1.850 × 10^−5^; OR = 1.034, 95% CI = 1.018–1.050). For every year of age, the odds of HPV positivity increased by 3%. In the extended set of 310 patients, a statistically significant association between age and HPV positivity could be shown, too (*p* = 0.006; OR = 1.030, 95% CI = 1.008–1.052).

In 310 patients, earlier use of hormonal IUD was associated with positivity of HPV (*p* = 0.033; OR = 3.211, 95% CI = 1.097–9.402), but not with CIN or CC.

The number of pregnancies and children, smoking, and obesity were not associated significantly with a positive HPV status.

## 4. Discussion

The estimated global HPV prevalence in humans is 11.7%. Sub-Saharan Africa (24.0%), Eastern Europe (21.4%), and Latin America (16.1%) showed the highest prevalence and Western Europe one of the lowest (9.0%). The HPV prevalence, as well as the burden of the HPV related diseases differ considerably by and within areas of countries according to several factors, including both extrinsic and intrinsic factors [15].

HPV is ranked as the second most common pathogen leading to gynecological disorders and the first in causing female cancers. Cervical cancer, which is the third most common female malignancy worldwide, is the most common disease resulting from infection with the 18 known HR HPVs (16, 18, 26, 31, 33, 35, 39, 45, 51, 52, 53, 56, 58, 59, 66, 68, 73, 82), followed by anal, vulvar, penile, vaginal, and oropharyngeal cancers [16]. The EUROArray HPV we used in this study is able to distinguish between these 18 HR HPVs as well as 12 LR HPVs by detecting the E6/E7 genes of HPV. The test was validated by the Valgent framework in 2018 [17]. Additionally, the EUROArray HPV was compared to a number of other commercial HPV assays, and showed good concordance for HR-HPV detection, with comparable sensitivity and specificity for ≥CIN 2 detection [18]. In a recent study, we compared the HPV genotype-specific concordance between EUROArray HPV and HPV 3.5 LCD-Array Kit that is suitable to distinguish HPV by the L1 genes, and found an agreement of 92.0% and a kappa value of 0.83 for detecting HR HPV [19].

Worldwide, the five most common HPV subtypes among women with normal cervical cytology were HPV 16 (3.2%), HPV 18 (1.4%), HPV 52 (0.9%), HPV 31 (0.8%), and HPV 58 (0.7%) while the corresponding prevalence for the European continent were HPV 16 (4.8%), followed by HPV 31 (2.3%), HPV 18 (0.9%), HPV 39 (0.8%), and 33 and 66 (0.6% each) [15]. In our investigation, the five most common HPV subtypes of all of the 487 cervical samples were HPV 16 (28.1%), followed by HPV 31 (10.7%), HPV 52 (7.4%), HPV 53 (7.0%), HPV 18 (5.5%), and HPV 39 (5.3%). The portions of HPV 31, HPV 18, and HPV 39 were nearly comparable with the study from Bruni et al. [15]. Considering that the participants of our study were visiting the dysplasia consultation, we had expected to find elevated percentages of HPV subtypes compared to cohorts of women with normal cytology. Interestingly, we found a comparably high percentage of HPV 52 than what is more often reported in cohorts of the African, Northern American, and Asian continent [15].

The most common HPV subtype in vulvar swabs was HPV 16 at 53.9% followed by HPV 53 (17.7%) and HPV 31 (15.8%). In a study by De Sanjose et al. including 587 VIN and 1709 vulvar cancer patients, the percentage for HPV 16 at 72.5% was the most common subtype. The next most frequent HPV of their study was HPV 33 (6.5%) and HPV 18 (4.6%) with percentages comparable to those found in our study (HPV 33 6.2% and HPV 18 4.6%) [4]. Similar results were published in a meta-analysis of 5015 cases of vulvar cancer and 2764 cases of VIN, where the predominant HR HPV type was HPV 16 (80.2%), followed by HPV 33 (8.3%) and HPV 18 (3.3%) [20]. The meta-analysis of Smith et al. found that among 2790 vulvar (1379 invasive, 1340 VIN 2/3, 71 VIN 1) and 315 vaginal cases, HPV 16 was the most common type in vulvar (29.3%) and vaginal (55.4%) cancers, VIN 2/3 (71.2%) and VaIN 2/3 (65.8%) [21].

Depending on the examined cohort, varying frequencies of HPV subtypes will be detected. In a meta-analysis, the overall HPV prevalence was 67.8%, 85.3% and 40.4% among 90 VIN 1, 1,061 VIN 2/3 and 1873 vulvar carcinomas, respectively; 100%, 90.1% and 69.9% among 107 VaIN 1, 191 VaIN 2/3 and 136 vaginal carcinomas; and 91.5%, 93.9% and 84.3% among 671 AIN 1, 609 AIN 2/3 and 955 anal carcinomas, respectively [22]. HPV 16 was found in more than 75% and HPV 18 in less than 10% of HPV-positive vulvar, vaginal and anal carcinomas [22]. Because of our mixed cohort including CIN 0/normal, VaIN 0/normal, VIN 0/normal and AIN 0/normal, we had lower portions of HPV 16 in all entities. The reported HPV 18 prevalence of less than 10% is concordant with our study. Further concordance with this study is that HPV 16 is more frequent and HPV 18 is less frequent in vaginal, vulvar and anal compared to cervical samples.

In a study by Insinga and coworkers—referring to a defined population (review of 22 U.S. studies) with cervical cancer and its pre-stages—the top two HPV types contributing to disease were HPV 16/66 (15.3%) in CIN 1, HPV 16/31 (61.9%) in CIN 2/3, and HPV 16/18 (79.2%) in CC [23]. In our study, we analyzed a percentage of 26.1 for the combination of HPV 16/66 in CIN 1 samples. Concordant to the study of Insinga, we found that in CIN 2/3 samples HPV 16 and HPV 31 were the two most common types with a percentage of 67.4%. In CC samples, the percentage of HPV 16/18 was slightly lower than in the study from Insinga with 70.8% in our study.

In line with other studies, we showed that the most predominant HPV in either disease is HPV 16, and we identified HPV 31 as the second most common HR HPV, especially in HSIL and CC [24,25]. Compared to the percentages of other studies we detected 10% more multiple HR HPV infections [24,26]. This may have been caused by the patients selected in our cohort.

In contrast to many other studies [27], we detected lower percentages of HPV 18. Only in the cervical samples HPV 18 ranked at the fifth position of frequency and in the other areas it was detected even less.

In our study, HPV 53 was frequently detected in all investigated areas with percentages ranging from 11.3% (cervix) to 18.9% (anus). A study by Gong et al detected 8.8% of HPV 53 in a total of 1259 patients with cervicitis (10.0%), LSIL (9.0%), HSIL (7.3%) and CC (0.0%) [28]. In our study, comparable percentages of HPV 53 could be identified in HSIL with 8.3%, whereas no HPV 53 was found in LSIL, but 8.3% HPV 53 in CC. Other studies also described frequent detection of the subtype HPV 53 [26,29,30]. Furthermore, HPV 53 is described to occur more likely in multiple infections together with other types [26]. In our study, 82.2% of HPV-53-positive cervical samples showed a multiple infection with at least one additional HR HPV. As an explanation, the low carcinogenicity reported for HPV 53 compared to other HR HPV, and therefore the tendency for an association with HR HPV subtypes could be reasonable.

Another aim of our study was to compare the HPV infection and subtypes of one patient in different anogenital areas and investigate a possible concordance of the different subtypes. As expected, we found the highest concordance in cervical, vaginal and vulvar samples, especially for the most common HPV subtypes HPV 16, HPV 31, and HPV 52. Another study group reported a similar prevalence for any carcinogenic HPV type in vaginal and cervical specimens compared to our study [31].

In addition to 16 and 31, less frequent HPV subtypes such as 51 and 73 that were detected more often in anal samples, displayed a substantial or moderate concordance with cervical, vaginal as well as vulvar samples. Similar results were reported by Nasioutziki and coworkers who showed that the cervical HPV infections of women with prevalent cervical dysplasia, correlated with the HPV types detected in their anal canal [32]. No or just light or fair concordance was detected between oral and other samples, suggestively due to the local distance of the low number of HPV infections.

Age-specific HPV distribution worldwide as well as in Western Europe presents with a first peak at younger ages [15,33], which is concordant with our findings that the highest infection rate is displayed by the younger women. In a veteran population of South Florida, the highest positive HR HPV test rates were found in the third and eighth decades of life at 25.1% and 22.0%, respectively [34]. However, the data regarding the eighth decade consisted of a small sample size. Concordant to the latter, our study found the highest peak at 30–39 years and a small peak at 70–79 years.

We showed that in a set of 310 patients earlier use of hormonal IUD was associated with positivity of HPV, but not with CIN or CC. Lekovich and coworker reported that hormone-containing IUDs could be associated with decreased HR HPV infection clearance and possibly increased acquisition compared to copper-containing IUDs [35]. In a population-based study, an association between HR HPV persistence that is obligatory for CC development and use of hormonal IUD in 7778 patients positive for HPV was not found [36].

A limitation of the study is the mixed cohort of patients visiting our dysplasia consultation resulting in the small number of samples from cancer patients and especially from patients with INs of entities other than cervical (VaIN, VIN, AIN).

## 5. Conclusions

The purpose of our study was to analyze a broad set of HR HPV and LR HPV in five areas for a better elucidation of HPV infections. This should add information on the spreading of infections between areas by analyzing co-infections with identical HPV subtypes. Furthermore, the correlation of infections with clinical data should provide further knowledge about incidences of HPV-attributable cancers and their precursors. We found, according to other studies, HPV 16 as the most common HPV and HPV 31 as second most in cervical and vaginal samples. HPV 53, which is not included in most other tests, was second most common in vulvar, anal and oral samples. HPV 53 is marked as possibly carcinogenic and is not well investigated. Our study found HPV 53 in CC and preferentially in HSIL samples. HPV 73 is an HR HPV and is also not covered by most other tests. We found HPV 73 with highest percentages in anal samples and with good concordance between the different areas. Further studies of these specific subtypes could be useful to clarify the importance of HPV 53 and HPV 73 in neoplasia areas next to the cervix.

In conclusion, the testing of many specific subtypes is valuable for the association of specific HPV subtypes with lesions and their precursors in various areas.

## Figures and Tables

**Figure 1 cancers-14-03136-f001:**
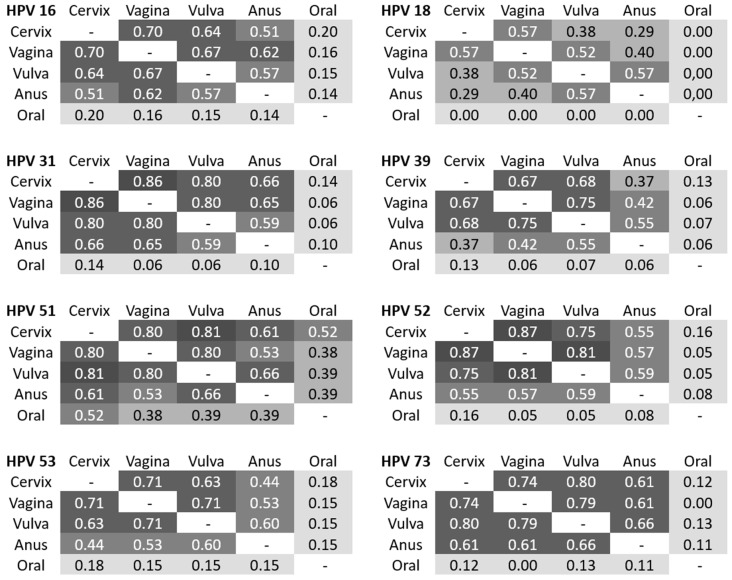
Concordance analyses by determination of Cohen’s kappa coefficients. Prevalence of the eight most frequent HR HPV (high risk human papilloma virus) types and comparison between the different anatomical areas. κ = 0.01–0.20 (slight concordance); κ = 0.21–0.40 (fair concordance); κ = 0.41–0.60 (moderate concordance); κ = 0.61–0.80 (substantial concordance); κ = 0.81–1.00 ((almost) perfect concordance).

**Figure 2 cancers-14-03136-f002:**
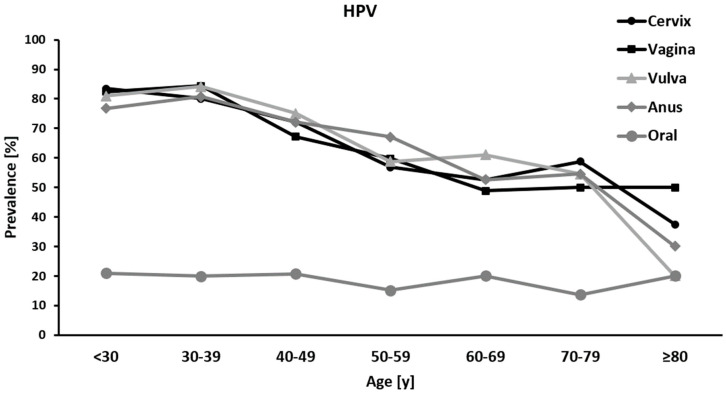
HPV distribution by age group. HPV = human papilloma virus.

**Table 1 cancers-14-03136-t001:** Patients’ demographics.

Characteristica	*n* (%)
Age (years)	total 508 *
15–29	91 (17.9)
30–39	141 (27.8)
40–49	116 (22.8)
50–59	87 (17.1)
60–69	41 (8.1)
70–79	22 (4.3)
≥80	10 (2.0)
Pregnancies	total 497 *
None	133 (26.8)
Once	119 (23.9)
≥twice	245 (49.3)
Children	total 497 *
0	195 (39.2)
1	119 (23.9)
2	128 (25.8)
≥3	55 (11.1)
Smoking	total 484 *
No	290 (59.9)
Yes	194 (40.1)
Adiposity (BMI > 30)	total 508 *
No	412 (81.1)
Yes	96 (18.9)

* Total numbers differ because of incomplete data. BMI = body mass index.

**Table 2 cancers-14-03136-t002:** Distribution of HPV samples of non-vaccinated and vaccinated patients.

	CC	CIN 3	CIN 2	CIN 1	VIN/Ca	AIN 1	PAP 3/4	PAP 2	Normal	Total	Total [%]
Vaccination (−/+)	−/+	−/+	−/+	−/+	−/+	−/+	−/+	−/+	−/+	−/+	−/+
*n* (−/+)	7/0	51/7	14/8	13/1	5/0	0/1	18/10	20/8	14/2	142/37	100/100
HPV 16	6/0	28/2	5/0	3/0	4/0	0/0	11/3	4/1	1/0	62/6	43.7/16.2
HPV 18	0/0	3/1	1/1	0/0	0/0	0/0	0/0	1/0	1/1	6/3	4.2/8.1
HPV 31	0/0	9/0	4/3	0/1	0/0	0/0	2/1	3/0	0/0	18/5	12.7/13.5
HPV 33	0/0	1/2	0/1	0/0	0/0	0/0	1/0	0/0	0/0	2/3	1.4/8.1
HPV 45	0/0	1/0	0/0	1/0	0/0	0/0	1/0	0/0	0/0	3/0	2.1/0
HPV 52	0/0	5/0	1/1	0/0	0/0	0/0	0/0	0/1	0/0	6/2	4.2/5.4
HPV 58	0/0	1/0	0/1	1/0	0/0	0/0	0/2	0/0	0/0	2/3	1.4/8.1
HR other	0/0	3/1	1/1	3/0	0/0	0/0	2/3	5/0	3/0	17/5	12.0/13.5
no HPV	1/0	0/1	2/0	5/0	1/0	0/1	1/1	7/6	9/1	26/10	18.3/27.0

**Table 3 cancers-14-03136-t003:** HPV prevalence and cytological examination.

Cytology	*n*	Anatomical Site	
		**Cervix**	
		HPV+	HR HPV+
		*n* (%)	*n* (%)
CIN 0/normal	288	168 (58.3)	129 (44.9)
CIN 1	23	17 (73.9)	16 (69.6)
CIN 2	42	41 (97.6)	39 (92.9)
CIN 3	90	87 (96.7)	85 (94.4)
CC	24	23 (95.8)	22 (91.7)
		**Vagina**	
		HPV +	HR HPV+
		*n* (%)	*n* (%)
VaIN 0/normal	500	352 (70.4)	310 (62.0)
VaIN 1	4	4 (100.0)	3 (75.0)
VaIN 2	1	1 (100.0)	1 (100.0)
VaIN 3	2	2 (100.0)	2 (100.0)
VaC	2	2 (100.0)	2 (100.0)
		**Vulva**	
		HPV +	HR HPV+
		*n* (%)	*n* (%)
VIN 0/normal	473	343 (73.4)	303 (64.1)
VIN 1	2	0 (0.0)	0 (0.0)
VIN 2	6	6 (100.0)	6 (100.0)
VIN 3	6	6 (100.0)	6 (100.0)
VC	15	66.7 (10)	8 (53.3)
		**Anus**	
		HPV+	HR HPV+
		*n* (%)	*n* (%)
AIN 0/normal	477	339 (71.1)	286 (60.0)
AIN 1	2	2 (100.)	2 (100.)
AIN 2	0	-	-
AIN 3	3	3 (100.0)	3 (100.0)
AC	0	-	-

CIN = cervical intraepithelial neoplasia, CC = cervical carcinoma, VaIN = vaginal intraepithelial neoplasia, VaC = vaginal carcinoma, VIN = vulvar intraepithelial neoplasia, VC = vulvar carcinoma, AIN = anal intraepithelial neoplasia, AC = anal carcinoma.

**Table 4 cancers-14-03136-t004:** Distribution of HR HPV genotypes in different areas. Percentages of single and multiple HR HPV are based on the number of HR HPV positive samples from the respective areas.

Area	Cervix	Vagina	Vulva	Anus	Oral
	*n* (%)	*n* (%)	*n* (%)	*n* (%)	*n* (%)
total	487	509	502	482	507
no HR HPV	187 (38.4)	191 (37.5)	179 (35.7)	191 (39.6)	433 (85.4)
HR HPV	300 (61.6)	318 (62.5)	323 (64.3)	291 (60.4)	74 (14.6)
single HR HPV	164 (54.7)	161 (50.6)	182 (56.3)	169 (58.1)	58 (78.4)
multiple HR HPV	136 (45.3)	157 (49.4)	141 (43.7)	122 (41.9)	16 (21.6)
2 HR HPV	82 (27.3)	90 (28.3)	78 (24.1)	69 (23.7)	14 (18.9)
3 HR HPV	36 (12.0)	43 (13.5)	40 (12.4)	29 (10.0)	1 (1.4)
4 HR HPV	11 (3.7)	13 (4.1)	15 (4.6)	15 (5.2)	1 (1.4)
5 HR HPV	6 (2.0)	7 (2.2)	3 (0.9)	4 (1.4)	0 (0)
>5 HR HPV	1 (0.3)	4 (1.3)	5 (1.5)	5 (1.7)	0 (0)

HR HPV = high-risk human papilloma virus.

**Table 5 cancers-14-03136-t005:** Frequencies of HPV genotypes detected in each area.

	Cervix	Vagina	Vulva	Anus	Oral
HR HPV	*n* (%) Positive	*n* (%) Positive	*n* (%) Positive	*n* (%) Positive	*n* (%) Positive
16	137 (45.7)	159 (50.0)	174 (53.9)	157 (54.0)	59 (79.7)
18	27 (9.0)	22 (6.9)	15 (4.6)	16 (5.5)	1 (1.4)
26	18 (6.0)	14 (4.4)	7 (2.2)	1 (0.3)	0 (0.0)
31	52 (17.3)	51 (16.0)	51 (15.8)	33 (11.3)	3 (4.1)
33	19 (6.3)	24 (7.6)	20 (6.2)	17 (5.8)	0 (0.0)
35	5 (1.7)	6 (1.9)	5 (1.6)	6 (2.1)	1 (1.4)
39	26 (8.7)	30 (9.4)	25 (7.7)	28 (9.6)	2 (2.7)
45	20 (6.7)	22 (6.9)	18 (5.6)	16 (5.5)	0 (0.0)
51	25 (8.3)	29 (9.1)	28 (8.7)	28 (9.6)	7 (9.5)
52	36 (12.0)	40 (12.6)	35 (10.8)	24 (10.3)	1 (1.4)
53	34 (11.3)	44 (13.8)	57 (17.7)	55 (18.9)	11 (14.9)
56	13 (4.3)	19 (6.0)	17 (5.3)	12 (4.1)	1 (1.4)
58	24 (8.0)	25 (7.9)	21 (6.5)	17 (5.8)	0 (0.0)
59	14 (4.7)	10 (3.1)	11 (3.4)	13 (4.5)	2 (2.7)
66	20 (6.7)	28 (8.8)	22 (6.8)	20 (6.9)	0 (0.0)
68	18 (6.0)	23 (7.2)	25 (7.7)	23 (7.9)	1 (1.4)
73	19 (6.3)	24 (7.6)	26 (8.1)	30 (10.3)	3 (4.1)
82	9 (3.0)	14 (4.4)	11 (3.4)	10 (3.4)	1 (1.4)
LR HPV					
6	25 (15.8)	32 (17.1)	32 (17.7)	43 (22.6)	7 (17.5)
11	3 (1.9)	4 (2.1)	3 (1.7)	5 (2.6)	0 (0.0)
40	36 (22.8)	30 (16.0)	20 (11.1)	3 (1.6)	0 (0.0)
42	46 (29.1)	56 (30.0)	61 (33.7)	57 (30.0)	24 (60.0)
43	9 (5.7)	13 (7.0)	18 (9.9)	16 (8.4)	0 (0.0)
44	18 (11.4)	24 (12.8)	24 (13.3)	29 (27.9)	0 (0.0)
54	39 (24.7)	46 (24.6)	48 (26.5)	53 (27.9)	7 (17.5)
61	15 (9.5)	30 (16.0)	33 (18.2)	31 (16.3)	4 (10.0)
70	10 (6.3)	14 (7.5)	7 (3.9)	7 (3.7)	0 (0.0)
72	1 (0.6)	2 (1.1)	3 (1.7)	2 (1.1)	1 (2.5)
81	7 (4.4)	11 (5.9)	11 (6.1)	11 (5.8)	1 (2.5)
89	8 (5.1)	18 (9.6)	17 (9.4)	12 (6.3)	1 (2.5)

HR HPV = high-risk human papilloma virus.

**Table 6 cancers-14-03136-t006:** Concordance [φ] between the HPV prevalence for the different sites.

Concordance	Cervix/Vagina		Cervix/Vulva		Cervix/Anus		Cervix/Oral	
	%	φ	%	φ	%	φ	%	φ
HPV	86.7	0.67	84.2	0.60	74.1	0.36	40.4	0.10
HR HPV	81.5	0.59	79.4	0.55	73.9	0.44	70.1	0.22
16	87.5	0.70	84	0.64	79.1	0.51	72.4	0.20
18	95.9	0.57	94.8	0.38	93.9	0.29	94.2	−0.01
26	95.7	0.26	96.5	0.28	96.3	-	96.3	-
31	97.3	0.86	96.2	0.80	94.3	0.66	89.5	0.14
33	99.2	0.90	97.5	0.66	96.7	0.56	96.1	-
35	99.8	0.91	99.6	0.80	99.3	0.73	98.8	−0.01
39	96.5	0.67	96.9	0.68	93.5	0.37	94.6	0.13
45	98.4	0.79	98.3	0.77	96.1	0.46	95.9	-
51	97.9	0.80	98.1	0.81	95.9	0.61	96.3	0.52
52	98.2	0.87	96.7	0.75	94.3	0.55	92.8	0.16
53	95.7	0.71	93.3	0.63	90	0.44	92.4	0.18
56	98.6	0.78	98.1	0.68	97.4	0.49	97.5	0.27
58	99.2	0.91	97.7	0.74	96.7	0.61	95.1	-
59	99.2	0.84	98.1	0.64	96.7	0.43	97.1	0.18
66	97.9	0.78	98.1	0.77	97.4	0.67	95.9	-
68	97.7	0.69	97.3	0.66	95.4	0.44	96.1	−0.01
73	97.7	0.74	98.1	0.80	95.9	0.61	95.9	0.12
82	99.2	0.83	99.8	0.95	98.5	0.63	97.9	−0.01
LR HPV	86.2	0.71	84.6	0.67	73.3	0.43	46.8	0.15
6	98.6	0.87	97.9	0.81	95.2	0.66	95.9	0.44
11	99.8	0.87	100	1.00	99.6	0.77	99.4	-
40	91.8	0.32	91.5	0.20	92.4	-0.01	92.6	-
42	95.3	0.75	93.5	0.68	91.7	0.58	89.7	0.25
43	98.4	0.64	98.1	0.70	97.2	0.49	98.1	-
44	98.4	0.80	96.5	0.55	95.4	0.52	96.3	-
54	94.9	0.67	94.2	0.64	91.5	0.52	91.8	0.16
61	96.7	0.63	95.2	0.50	94.8	0.42	97.1	0.29
70	99.2	0.84	99	0.71	98.7	0.62	97.9	-
72	100	1.00	99.6	0.58	99.3	0.00	100	1.00
81	99.6	0.88	98.5	0.59	99.3	0.83	98.8	0.38
89	98.2	0.68	98.3	0.70	98.5	0.63	98.6	0.35

HR HPV = high-risk human papilloma virus, LR HPV = low-risk HPV.

**Table 7 cancers-14-03136-t007:** Prevalence of HPV subtypes in CC and CIN. Prevalence of HR HPV subtypes indicated by percentages of either number of CC, CIN 3, CIN 2, CIN 1 and CIN 0/normal; percentages in brackets (%).

	CIN 0/Normal	CIN 1	CIN 2	CIN 3	CC
	*n* (%)	*n* (%)	*n* (%)	*n* (%)	*n* (%)
total	288	23	42	90	24
HPV+	168 (58.3)	17 (73.9)	41 (97.6)	87 (96.7)	23 (95.8)
HR HPV+	129 (44.8)	16 (69.6)	39 (92.9)	85 (94.4)	22 (91.7)
16	52 (18.1)	5 (21.7)	14 (33.3)	44 (48.9)	16 (66.7)
18	15 (5.2)	-	5 (11.9)	6 (6.7)	1 (4.2)
26	15 (5.2)	2 (8.7)	1 (2.4)	-	-
31	18 (6.3)	3 (13.0)	10 (23.8)	21 (23.3)	-
33	3 (1.0)	-	5 (11.9)	9 (10.0)	1 (4.2)
35	3 (1.0)	-	-	2 (2.2)	-
39	19 (6.6)	-	2 (4.8)	5 (5.6)	-
45	6 (2.1)	1 (4.3)	3 (7.1)	9 (10.0)	1 (4.2)
51	14 (4.9)	-	3 (7.1)	8 (8.9)	-
52	8 (2.8)	2 (8.7)	9 (21.4)	15 (16.7)	2 (8.3)
53	19 (6.6)	-	4 (9.5)	7 (7.8)	2 (8.3)
56	6 (2.1)	-	3 (7.1)	4 (4.4)	-
58	9 (3.1)	2 (8.7)	4 (9.5)	7 (7.8)	2 (8.3)
59	7 (2.4)	2 (8.7)	1 (2.4)	4 (4.4)	-
66	12 (4.2)	1 (4.3)	4 (9.5)	2 (2.2)	1 (4.2)
68	8 (2.8)	2 (8.7)	4 (9.5)	4 (4.4)	-
73	8 (2.8)	1 (4.3)	5 (11.9)	3 (3.3)	2 (8.3)
82	3 (1.0)	1 (4.3)	3 (7.1)	3 (3.3)	-

HR HPV = high-risk human papilloma virus, CIN = cervical intraepithelial neoplasia, CC = cervical carcinoma.

**Table 8 cancers-14-03136-t008:** Prevalence of HPV, HR HPV, 16, 31, and 52 in HSIL and LSIL samples.

	LSIL	HSIL	*p*
Number [*n*]	23	132	
HPV+ [%]	73.9	97.0	2.213 × 10^−5^ *
HR HPV+ [%]	69.6	93.9	2.150 × 10^−4^ *
HPV 16 [%]	21.7	43.9	0.046 *
HPV 31 [%]	13.0	23.5	0.267
HPV 52 [%]	8.7	18.2	0.264

HR HPV = high-risk human papilloma virus, LSIL = low squamous intraepithelial lesion, HSIL = high squamous intraepithelial lesion; * *p* ≤ 0.05.

## Data Availability

The data presented in this study are available on request from the corresponding author.

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
