# Peer review of "Distribution of HPV Subtypes in Diverse Anogenital and Oral Samples from Women and Correlation of Infections with Neoplasia of the Cervix"

_cancers, 2022, doi:10.3390/cancers14133136_

Round 1

Reviewer 1 Report

Braeutigam et al. habe presented a very interesting study investigating HPV suptypes in tissue from anogenital and oral samples.

The study ist well presented and described. My only concern is that the conclusion section should be shortened.

Author Response

Answer to Reviewer 1 Comments

Reviewer 1: Comments and Suggestions for Authors

Braeutigam et al. have presented a very interesting study investigating HPV suptypes in tissue from anogenital and oral samples.

The study is well presented and described. My only concern is that the conclusion section should be shortened.

Thank you very much for your review and your valuable comment! All changes to the manuscript were performed in red color.

The conclusion section was shortened

  1. Conclusions

The purpose of our study was to analyze a broad set of HR HPV and LR HPV in five areas for a better elucidation of HPV infections. This should add information on the spreading of infections between areas by analyzing co-infections with identical HPV subtypes. Furthermore, the correlation of infections with clinical data should give further knowledge about incidences of HPV-attributable cancers and their precursors. We found, according to other studies, HPV 16 as the most common HPV and HPV 31 as second most in cervical and vaginal samples. HPV 53, which is not included in most other tests, was second most common in vulvar, anal and oral samples. HPV 53 is marked as possibly carcinogenic and is not well investigated. Our study found HPV 53 in CC and preferentially in HSIL samples. HPV 73 is a HR HPV and not covered by most of other tests as well. We found HPV 73 with highest percentages in anal samples and with good concordances between the different areas. Further studies for these specific subtypes could be useful to clarify the importance of HPV 53 and HPV 73 in neoplasia areas next to the cervix.

In conclusion, the testing of many specific subtypes is valuable for the association of specific HPV subtypes with lesions and their precursors in various areas.

Reviewer 2 Report

In this study, the distribution of HPV subtypes in diverse anogenital and oral samples  correlation of infections with neoplasia were analyzed in patients from a dysplasia clinic. The study is sound and complete in methods and results. The  number of samples are not large but it is a minor concern.

comments

The authors note that the EUROArray test can discriminate 18 HR and 12 LR HPV. they should add a note of comparison to other methods. 

How many samples had co-infections with both HR and LR at these anatomical locations? If so what were the types?

The authors mention that vaccination rates were low. What percent of vaccinated individuals had cytological abnormalities? How many doses/ what vaccines were taken by these individuals? Did they have  HPV types not covered by the vaccine?

It is very likely for the samples to be contaminated from adjacent anatomical sites in the case of cervix, vagina and vulva. Please elaborate on the methods of sample collection? Were all the anatomical sites  swabbed at the same time? 

Are there other published studies carried out using the euroarray platform? If so, how do these studies compare? 

Author Response

Answer to Reviewer 2 Comments

Reviewer 2: Comments and Suggestions for Authors

In this study, the distribution of HPV subtypes in diverse anogenital and oral samples correlation of infections with neoplasia were analyzed in patients from a dysplasia clinic. The study is sound and complete in methods and results. The number of samples are not large but it is a minor concern.

Comments

Thank you very much for your review and your very valuable comments! All changes to the manuscript were performed in red color.

The authors note that the EUROArray test can discriminate 18 HR and 12 LR HPV. they should add a note of comparison to other methods.

On page 2 in the introduction section we add:

In clinical routine diagnostics, detection of HPV is discriminated in only the two distinguished HPV 16 and HPV 18 while 12 HR HPV are pooled in the group of “HR HPV others” (31, 33, 35, 39, 45, 51, 52, 56, 58, 59, 66, 68).

How many samples had co-infections with both HR and LR at these anatomical locations? If so what were the types?

On page 8, result section 3.3 we add:

Looking for co-infections of both HR and LR HPV in the same anatomical areas, we found 109 (22.4%) in the cervix, 144 (28.3%) in the vagina, 139 (27.7%) in the vulva, 137 (28.4%) in the anus, and 17 (3.4%) in the oral cavity. HR HPV 16 showed the highest co-infection rates together with the LR HPV 42 with rising percentages from cervix (13.8%), vagina (14.6%), vulva (17.3%), anus (21.2%), to the oral cavity (29.4%). HR HPV 31 was analyzed most frequently in combination with the LR HPV 54 with decreasing percentages from cervix (8.3%), vagina (7.6%), vulva (7.2%), anus (3.7%), to the oral cavity (0%).

The authors mention that vaccination rates were low. What percent of vaccinated individuals had cytological abnormalities? How many doses/ what vaccines were taken by these individuals? Did they have HPV types not covered by the vaccine?

On page 4 we add in the results section 3.1 a text section and on page 5 table 2

HPV vaccination was sparse and not sufficient for deep statistical analyses; 37 women of 181 (20.4%) reported being immunized. As expected, because in Germany the vaccination program started in 2007, the median age of vaccinated women was with 28.3 ± 7.3 years younger than the median age of the non-vaccinated group (39.7 ± 10.9 years). Furthermore, the information about the sort of vaccine was incomplete. Complete vaccination was reported for 13 patients with one applied Gardasil 4, four Gardasil 9, and two Cervarix. For the others neither vaccine type nor the frequency of vaccination was known. In total, within the 37 vaccinated patients ten had a normal cytology, ten Pap3, one AIN 1, one CIN 1, eight CIN 2, and seven CIN 3. Of the vaccinated patients with diagnoses of CIN 1 – 3, all were HR HPV-positive. Of these HR HPV, six were HPV 16 or HPV 18 which would have been covered by all vaccines and ten were HR HPV subtypes covered only by Gardasil 9. Samples with Pap 3 or normal cytology showed less HPV 16, 18 or HR HPV subtypes. Of the 142 patients without vaccination, seven were diagnosed with CC, 51 with CIN 3, 14 with CIN 2, 13 with CIN 1, 18 with a conspicuous Pap, and 44 with Pap 2a/normal cytology. In six of seven CC samples HPV 16 was analyzed and would have been covered by any vaccination type. In CIN 3 samples 48 of 51 samples showed HR HPV 16, 18, 31, 33, 45, or 52 and the patients would have benefitted from vaccination with Gardasil 9; and still 32 samples with HPV 16 or 18 would have benefitted from vaccination with either Gardasil 4 or Cervarix. Contribution of the seven HR HPV included in the nonavalent vaccination with Gardasil 9 in samples of non-vaccinated and vaccinated were shown in Table 2.

Table 2. Distribution of HPV samples of non-vaccinated and vaccinated patients

CC

CIN 3

CIN 2

CIN 1

VIN/Ca

AIN 1

PAP 3/4

PAP 2

normal

total

total [%]

vaccination (-/+)

-/+

-/+

-/+

-/+

-/+

-/+

-/+

-/+

-/+

- /+

- /+

n (- /+)

7/0

51/7

14/8

13/1

5/0

0/1

18/10

20/8

14/2

142/37

100/100

HPV 16

6/0

28/2

5/0

3/0

4/0

0/0

11/3

4/1

1/0

62/6

43.7/16.2

HPV 18

0/0

3/1

1/1

0/0

0/0

0/0

0/0

1/0

1/1

6/3

4.2/8.1

HPV 31

0/0

9/0

4/3

0/1

0/0

0/0

2/1

3/0

0/0

18/5

12.7/13.5

HPV 33

0/0

1/2

0/1

0/0

0/0

0/0

1/0

0/0

0/0

2/3

1.4/8.1

HPV 45

0/0

1/0

0/0

1/0

0/0

0/0

1/0

0/0

0/0

3/0

2.1/0

HPV 52

0/0

5/0

1/1

0/0

0/0

0/0

0/0

0/1

0/0

6/2

4.2/5.4

HPV 58

0/0

1/0

0/1

1/0

0/0

0/0

0/2

0/0

0/0

2/3

1.4/8.1

HR other

0/0

3/1

1/1

3/0

0/0

0/0

2/3

5/0

3/0

17/5

12.0/13.5

no HPV

1/0

0/1

2/0

5/0

1/0

0/1

1/1

7/6

9/1

26/10

18.3/27.0

It is very likely for the samples to be contaminated from adjacent anatomical sites in the case of cervix, vagina and vulva. Please elaborate on the methods of sample collection? Were all the anatomical sites swabbed at the same time?

On page 2 in the Materials and Methods section 2.1 we add:

For sampling, samples were taken of the five anatomical areas (cervix, vagina, vulva (between major and minor labia), anus (perianal), and oral cavity (cheek)) at the same time by a physician using flocked swabs in eNat® (Guanidine-thiocyanate based) medium (Copan Italia, Brescia, Italy) and stored at room temperature.

Are there other published studies carried out using the euroarray platform? If so, how do these studies compare?

On page 13 in the conclusion section we add:

The EUROArray HPV we used in this study is able to distinguish between these 18 HR HPV as well as 12 LR HPV by detecting the E6/E7 genes of HPV. The test was validated by the Valgent framework in 2018 [18]. Additionally, the EUROArray HPV was compared to a number of other commercial HPV assays, and showed good concordance for HR-HPV detection, with comparable sensitivity and specificity for ≥ CIN2 detection [19]. In a recent study we compared the HPV genotype-specific concordance between EUROArray HPV and HPV 3.5 LCD-Array Kit that is suitable to distinguish HPV by the L1 genes, and found an agreement of 92.0% and a kappa value of 0.83 for detecting HR HPV [20].   

Reviewer 3 Report

Brautigam et al report the distribution of HPV types in a cohort of anogenital and oral samples obtained from a German centre over the last 5 years.  HPV detection was performed using the relatively new EUROArray test on 509 women.  Lots of appropriate comparisons are done between HPV status, clinical characteristics and anatomical location.

This is a generally well written manuscript, with lots of the usual comparisons. The data line up with observations in many other cohorts.  Not a lot new here, except perhaps the data on HPV53 and 73, which are not often reported, but are detected by the EUROArray platform.

Minor comments that could be addressed.

1) unfortunately, vaccine status was "sparsely" reported with information for only 181 members of the cohort.  Only 37 of these reported vaccination.  I recommend that the authors try to compare those 37 to the other 144 others with known lack of vaccination to see if any evidence of protection/HPV type replacement from specific types is observed.  Presumably these were all younger individuals and a table comparing the characteristics of these 181 people could be added.  Do they know which vaccine was administered?

2) the overweight characteristic should be added to the demographics shown in table 1.

Minor typos/corrections:

-second paragraph of results: shoud be "severely overweight" not "severe overweight"

-first paragraph after table 6:  add ref for "new nomenclature"

-second paragraph after table 7: hormone coil should be replaced with "hormonal intrauterine device" to be consistent with other description

-third paragraph before conclusion: need a reference for the South Florida study and some restructuring of the sentence to make the point clearer.

- in the conclusions, replace "preferable in HSIL samples" with "are found preferentially in HSIL samples"

Author Response

Answer to Reviewer 3 Comments

Reviewer 3: Comments and Suggestions for Authors

Brautigam et al report the distribution of HPV types in a cohort of anogenital and oral samples obtained from a German centre over the last 5 years.  HPV detection was performed using the relatively new EUROArray test on 509 women.  Lots of appropriate comparisons are done between HPV status, clinical characteristics and anatomical location.

This is a generally well written manuscript, with lots of the usual comparisons. The data line up with observations in many other cohorts.  Not a lot new here, except perhaps the data on HPV53 and 73, which are not often reported, but are detected by the EUROArray platform.

Thank you very much for your review and your very valuable comments! All changes to the manuscript were performed in red color.

Minor comments that could be addressed.

  • unfortunately, vaccine status was "sparsely" reported with information for only 181 members of the cohort. Only 37 of these reported vaccination.  I recommend that the authors try to compare those 37 to the other 144 others with known lack of vaccination to see if any evidence of protection/HPV type replacement from specific types is observed.  Presumably these were all younger individuals and a table comparing the characteristics of these 181 people could be added.  Do they know which vaccine was administered?

On page 4 we add in the results section 3.1 a text section and on page 5 table 2

HPV vaccination was sparse and not sufficient for deep statistical analyses; 37 women of 181 (20.4%) reported being immunized. As expected, because in Germany the vaccination program started in 2007, the median age of vaccinated women was with 28.3 ± 7.3 years younger than the median age of the non-vaccinated group (39.7 ± 10.9 years). Furthermore, the information about the sort of vaccine was incomplete. Complete vaccination was reported for 13 patients with one applied Gardasil 4, four Gardasil 9, and two Cervarix. For the others neither vaccine type nor the frequency of vaccination was known. In total, within the 37 vaccinated patients ten had a normal cytology, ten Pap3, one AIN 1, one CIN 1, eight CIN 2, and seven CIN 3. Of the vaccinated patients with diagnoses of CIN 1 – 3, all were HR HPV-positive. Of these HR HPV, six were HPV 16 or HPV 18 which would have been covered by all vaccines and ten were HR HPV subtypes covered only by Gardasil 9. Samples with Pap 3 or normal cytology showed less HPV 16, 18 or HR HPV subtypes. Of the 142 patients without vaccination, seven were diagnosed with CC, 51 with CIN 3, 14 with CIN 2, 13 with CIN 1, 18 with a conspicuous Pap, and 44 with Pap 2a/normal cytology. In six of seven CC samples HPV 16 was analyzed and would have been covered by any vaccination type. In CIN 3 samples 48 of 51 samples showed HR HPV 16, 18, 31, 33, 45, or 52 and the patients would have benefitted from vaccination with Gardasil 9; and still 32 samples with HPV 16 or 18 would have benefitted from vaccination with either Gardasil 4 or Cervarix. Contribution of the seven HR HPV included in the nonavalent vaccination with Gardasil 9 in samples of non-vaccinated and vaccinated were shown in Table 2.

Table 2. Distribution of HPV samples of non-vaccinated and vaccinated patients

CC

CIN 3

CIN 2

CIN 1

VIN/Ca

AIN 1

PAP 3/4

PAP 2

normal

total

total [%]

vaccination (-/+)

-/+

-/+

-/+

-/+

-/+

-/+

-/+

-/+

-/+

- /+

- /+

n (- /+)

7/0

51/7

14/8

13/1

5/0

0/1

18/10

20/8

14/2

142/37

100/100

HPV 16

6/0

28/2

5/0

3/0

4/0

0/0

11/3

4/1

1/0

62/6

43.7/16.2

HPV 18

0/0

3/1

1/1

0/0

0/0

0/0

0/0

1/0

1/1

6/3

4.2/8.1

HPV 31

0/0

9/0

4/3

0/1

0/0

0/0

2/1

3/0

0/0

18/5

12.7/13.5

HPV 33

0/0

1/2

0/1

0/0

0/0

0/0

1/0

0/0

0/0

2/3

1.4/8.1

HPV 45

0/0

1/0

0/0

1/0

0/0

0/0

1/0

0/0

0/0

3/0

2.1/0

HPV 52

0/0

5/0

1/1

0/0

0/0

0/0

0/0

0/1

0/0

6/2

4.2/5.4

HPV 58

0/0

1/0

0/1

1/0

0/0

0/0

0/2

0/0

0/0

2/3

1.4/8.1

HR other

0/0

3/1

1/1

3/0

0/0

0/0

2/3

5/0

3/0

17/5

12.0/13.5

no HPV

1/0

0/1

2/0

5/0

1/0

0/1

1/1

7/6

9/1

26/10

18.3/27.0

  • the overweight characteristic should be added to the demographics shown in table 1.

On page 4 we add

Adiposity (BMI>30)

total 508*

No

412 (81.1)

Yes

96 (18.9)

to the table 1 in the result section 3.1

Minor typos/corrections:

-second paragraph of results: shoud be "severely overweight" not "severe overweight"

On page 3 we changed into “severely overweight (adiposity; BMI>30).”

-first paragraph after table 6:  add ref for "new nomenclature"

On page 11 reference was added: new nomenclature [6]

-second paragraph after table 7: hormone coil should be replaced with "hormonal intrauterine device" to be consistent with other description

On page 12 hormone coil was replaced with “hormonal IUD”

-third paragraph before conclusion: need a reference for the South Florida study and some restructuring of the sentence to make the point clearer.

On page 14 the text part and reference were changed

In a veteran population of South Florida, the highest positive HR HPV test rates were found in the third and eighth decades of life with 25.1% and 22.0%, respectively [35].

- in the conclusions, replace "preferable in HSIL samples" with "are found preferentially in HSIL samples"

On page 15 preferable was replaced by and preferentially in HSIL samples
